# Phytochemicals Prime RIG-I Signaling and Th1-Leaning Responses in Human Monocyte-Derived Dendritic Cells

**DOI:** 10.3390/nu17223539

**Published:** 2025-11-12

**Authors:** Kaho Ohki, Takumi Iwasawa, Kazunori Kato

**Affiliations:** 1Department of Nutrition Sciences, Graduate School of Health and Sports Sciences, Toyo University, Tokyo 115-8650, Japan; s4h202300027@toyo.jp (K.O.); kzkatou@juntendo.ac.jp (K.K.); 2Institution of Life Innovation Studies, Toyo University, Tokyo 115-8650, Japan; 3Shizuoka Medical Research Center for Disaster, Juntendo University Shizuoka Hospital, Shizuoka 410-2295, Japan; 4Department of Nutrition Sciences, Faculty of Health and Sports Sciences, Toyo University, Tokyo 115-8650, Japan

**Keywords:** monocyte-derived dendritic cells, phytochemical, α-Mangostin, interferon, pattern recognition receptors, retinoic acid-inducible gene-I

## Abstract

**Background/Objective**: Dendritic cells (DCs) act as sentinels bridging innate and adaptive immunity, and their functions are strongly influenced by dietary and environmental factors. Phytochemicals such as α-Mangostin (A phytochemical, a xanthone derivative from Garcinia mangostina, known for its anti-inflammatory and antioxidant properties) are widely recognized for their antioxidant and anti-inflammatory effects, but their potential to modulate antiviral pattern recognition pathways remains unclear. This study investigated whether phytochemicals activate retinoic acid–inducible gene I (*RIG-I: DDX58*, a cytosolic receptor recognizing viral RNA and inducing antiviral responses)–dependent signaling in human monocyte-derived dendritic cells (MoDCs) and affect downstream T cell responses. **Methods**: MoDCs were generated from peripheral blood and stimulated with selected phytochemicals. RIG-I pathway–related transcripts were quantified by qPCR, and protein expression was assessed by Western blotting, intracellular flow cytometry, and immunofluorescence staining. Functional outcomes were evaluated by co-culturing MoDCs with T cells, followed by phenotypic analysis via flow cytometry and measurement of IFN-γ production by ELISA. **Results**: α-Mangostin stimulation increased *RIG-I (DDX58)* mRNA levels in MoDCs and induced time-dependent changes in intracellular protein expression. In co-culture, α-Mangostin–treated MoDCs tended to increase the proportion of OX40^+^ 4-1BB^+^ CD4^+^ T cells, accompanied by a significant elevation of IFN-γ levels in supernatants. Experiments with CpG-ODN (synthetic oligodeoxynucleotides mimicking bacterial DNA that activate TLR9) suggested context-dependent crosstalk between the TLR9 and RIG-I signaling axes. **Conclusions**: Phytochemicals, exemplified by α-Mangostin, prime antiviral responses in human DCs through upregulation of RIG-I and promote Th1-dependent immune responses. These findings suggest that phytochemicals may represent promising nutritional strategies to enhance antiviral immunity while mitigating excessive inflammation under infectious conditions.

## 1. Introduction

The skin is the first protective barrier against invading pathogens, and the mucous membranes that make up the inner walls of the nose and throat also function as barriers. However, the innate immune system, an intracellular mechanism against pathogens that breach these barriers, functions as a secondary barrier. Within this system, dendritic cells (DCs), discovered in 1973 [1], serve as pivotal sentinels linking innate and adaptive immunity [2]. Among the pattern recognition receptors (PRRs) expressed by dendritic cells (DCs), retinoic acid-inducible gene I (RIG-I) and Toll-like receptor 9 (TLR9) regulate antiviral immune responses through distinct intracellular localizations and recognition mechanisms. RIG-I functions as a cytoplasmic RNA sensor and recognizes viral double-stranded RNA and single-stranded RNA containing 5′-triphosphate moieties, thereby detecting a broad range of RNA viruses. Upon recognition, RIG-I activates the downstream MAVS-dependent signaling cascade, inducing the production of type I interferons and proinflammatory cytokines. In contrast, TLR9 localizes to endosomal compartments and recognizes unmethylated CpG motifs derived from DNA viruses. TLR9 signaling is transduced via the MyD88 pathway and functions in specific immune cell populations. Signals mediated through these receptors promote DC maturation and antigen presentation capacity, bridging innate and adaptive immunity through efficient induction of Th1 responses and cytotoxic T cells. Thus, RIG-I and TLR9 play central roles in orchestrating effective antiviral immune responses while appropriately regulating inflammation [3,4]. Type I interferons (IFNs), primarily represented by IFN-α and IFN-β, constitute a family of cytokines that play a central role in host defense against viral infections. They are rapidly produced upon recognition by pattern recognition receptors (PRRs), such as Toll-like receptors (TLRs)—a class of membrane-bound receptors that detect extracellular or endosomal microbial components including viral RNA and bacterial DNA—and RIG-I-like receptors (RLRs) [5,6], which recognize pathogen-associated molecular patterns (PAMPs). Beyond their direct antiviral effects, type I IFNs shape adaptive immunity by enhancing antigen presentation, promoting dendritic cell (DC) activation and maturation, and supporting T cell and B cell responses, thereby inducing IFN-γ, a type II interferon. Beyond their protective role in viral infections, type I IFNs are also involved in the regulation of inflammatory and autoimmune processes [7,8]. Furthermore, type I IFNs hold significant clinical importance as therapeutic agents, with recombinant IFN-α and IFN-β widely used in the treatment of viral infections and immune-mediated diseases. Currently, modulation of IFN signaling is being actively investigated for its antitumor effects [9,10] and applications in cancer immunotherapy [11,12].

DCs are classified into various subsets [13,14], among which plasmacytoid dendritic cells (pDCs) represent a distinctive subset characterised by their secretion of high levels of type I interferons [15]. Through rapid and robust type I IFN secretion [16], pDCs establish an antiviral state and orchestrate the activation of natural killer cells, B cells, and T cells [17]. Research on probiotics focusing on the antiviral activity of pDCs is also increasing [18,19]. Monocyte-derived dendritic cells (MoDCs) are frequently employed as a model to investigate the effects of environmental and nutritional factors on immune signaling [20].

Plant-derived natural compounds (phytochemicals) possess diverse chemical structures—including phenolic compounds, polyphenols, flavonoids, coumarins, terpenoids, and alkaloids—and these structural differences greatly influence their bioactivities and immunomodulatory properties. These compounds are known to exert a wide range of biological activities, such as antioxidant, anti-inflammatory, and antitumor effects, and numerous studies have reported their potential contributions to disease prevention and health maintenance [21,22,23,24,25]. In recent years, these natural compounds have attracted attention not only as mere metabolic regulators but also as immunomodulatory molecules that directly act on immune responses. In fact, compounds such as green tea catechins, resveratrol, sulforaphane, and β-glucan have been reported to enhance the cytotoxic activity of natural killer (NK) cells and to regulate cytokine production and differentiation of T cells [26,27,28,29,30], and their relationship to tumor immunity and infection defense has been discussed. Although plant-derived natural compounds exhibit a wide variety of chemical scaffolds and modes of action, and have been shown to affect intracellular signaling pathways and cellular functions, current knowledge regarding their effects on dendritic cells (DCs), which bridge innate and adaptive immunity, remains limited compared to the growing body of research on NK cells and T cells [31,32,33,34,35].

Since DCs play a central role in the activity of T cell responses via antigen presentation, immunomodulation targeting DCs may directly lead to the development of new therapeutic strategies. Therefore, elucidation of the effects of plant-derived natural components on DC function will contribute to understanding the molecular basis of immune response regulation and to the development of a wide range of clinical applications, including cancer immunotherapy, prevention of infectious diseases, and control of autoimmune diseases [36,37].

In this field, it remains unclear whether phytochemical stimulation enhances RIG-I-dependent signaling in monocyte-derived DCs, thereby potentially promoting type I interferon responses. Elucidating these interactions could yield novel insights into how dietary components contribute to host defence and potentially suggest new nutritional strategies to support immune resilience against viral infection. Therefore, in this study, we aimed to analyze whether phytochemical stimulation enhances RIG-I–dependent signaling in MoDCs and whether such priming promotes type I IFN–related outputs and Th1-skewed adaptive responses.

## 2. Materials and Methods

### 2.1. Cell Culture Medium

The culture medium for human MoDCs consists of RPMI 1640 medium (Nacalai Tesque, Kyoto, Japan) supplemented with 1 mM sodium pyruvate (Invitrogen, Waltham, MA, USA), 2.5 mM HEPES (Invitrogen), 100 U/mL penicillin-streptomycin (Invitrogen), 2 mM Gluta MAX (Invitrogen), 10% FBS (Biowest, Nuaillé, France), 50 μM 2-mercaptoethanol (2-ME) (BioRad, Hercules, CA, USA).

### 2.2. Donor Recruitment and Blood Collection

Peripheral blood was collected from *N* = 5 healthy adult volunteers (age 24–35; (2) female, (3) male) recruited at [Institution/Department], [Tokyo, Japan] during [May 2025–March 2026]. All donors provided written informed consent, and the study was approved by the Toyo University (protocol [2025-2HS]). Blood was drawn by venipuncture into Heparin tubes and processed within [≤2] h for PBMC isolation.

### 2.3. PBMC Isolation

Peripheral blood was obtained from healthy volunteers. PBMCs were isolated by density gradient centrifugation using a 1.077 g/mL polysucrose–sodium diatrizoate medium (Lymphocyte Separation Solution, Nacalai Tesque). Whole blood was diluted 1:1 with PBS containing 1 mM EDTA, carefully layered over the separation medium, and centrifuged at 400× *g* for 30 min at room temperature (brake off). The mononuclear cell layer at the plasma–medium interface was collected, washed twice with PBS, and resuspended in complete medium for MoDCs generation.

### 2.4. MoDCs Culture

MoDCs generated from peripheral blood mononuclear cells (PBMCs) were used for evaluating cytokine production and measuring Th differentiation-inducing capacity in co-culture with allogeneic CD4^+^ T cells. Peripheral blood mononuclear cells (PBMCs) were seeded into plates and allowed to stand under a 37 °C, 5% CO_2_ incubator for at least 1 h. After removal of non-adherent cells, adherent monocytes were cultured in complete medium supplemented with recombinant human GM-CSF (5 ng/mL; BioLegend, San Diego, CA, USA) and recombinant human IL-4 (10 ng/mL; BioLegend). On day 6, nonadherent undifferentiated MoDCs were collected, counted, and placed in a downstream for assay. On day 6, non-adherent immature MoDCs were collected and seeded into 12-well plates at 2 × 10^5^ cells/mL. On day 7, phytochemicals were added as indicated. The generation of GM-CSF- and IL-4-stimulated human monocyte-derived dendritic cells has been described previously [38].

### 2.5. Phytochemicals

MoDCs were stimulated with the following phytochemicals: Alpha-Mangostin (α-Mangostin) (Med Chem Express, Monmouth Junction, NJ, USA), Hesperetin (Fujifilm Wako, Osaka, Japan), Hesperidin (Fujifilm Wako), Neo hesperidin (Fujifilm Wako), Quercetin (Sigma-Aldrich, St. Louis, MO, USA), and Tangeretin (Sigma-Aldrich) Auraptene (Sigma-Aldrich). Purified analytical standards (≥95%) chemically identical to the naturally occurring molecules were used and adjusted to 10 mM using CultureSure^®^ DMSO (Fujifilm Wako).

We selected this panel to cover diverse scaffolds with reported immune relevance and dietary exposure: a xanthone (α-Mangostin; *Garcinia mangostana*), flavanones (hesperidin/hesperetin/neohesperidin; citrus peel), coumarins (auraptene), a polymethoxyflavone (tangeretin; citrus peel), and a flavonol (quercetin; widely present in onion, apple, tea). Concentration gradient assays (0.5–40 μM; 6–48 h) were performed for each phytochemical, monitoring BW5147 (provided by the JCRB Cell Bank) survival (WST-8). The working concentration was set at 5 µM, corresponding to the lowest concentration achieving an IC_50_ for α-Mangostin.

### 2.6. Antibodies

MoDCs were stained with the following fluorescently labelled anti-human antibodies: CD14-PE (63D3) (BioLegend), CD1c-APC/Cyanine7 (L161) (BioLegend), CD11c-PerCP/Cyanine5.5 (3.9) (BioLegend), HLA-DR-APC (G46-6) (BD Biosciences, CA, USA).

Plasmacytoid dendritic cells were fractionated by CD1c^−^, HRA-DR^+^, CD11c^−^, CD304^+^, Lineage PE-cocktail^−^, and CD123^+^ after staining with the following fluorescently labeled anti-human antibodies: CD1c-APC/Cyanine7 (L161) (BioLegend), HLA-DR-APC (G46-6) (BD Biosciences), CD304-PE/Cyanine7 (12C2) (BioLegend), CD11c-PerCP/Cyanine5.5 (3.9) (BioLegend), PE-conjugated Lineage cocktail (CD14-PE (63D3) (BioLegend), CD3-PE (OKT3) (BioLegend), CD16-PE (B73.1) (BD Biosciences), CD19-PE (H1B19) (BioLegend), CD34-PE (IMMUNOTECH, Marseille, France), CD56-PE (B159) (BD Biosciences), CD123-Alexa Fluor488 (6H6) (BioLegend).

T cells were fractionated by CD56^−^, CD2^+^ after staining with the following fluorescently labeled anti-human antibodies: CD56-PerCP/Cyanine5.5 (HCD56) (BioLegend), CD2-FITC (TS1/8) (BioLegend). T cell isolation was performed according to previously reported methods [39]. Furthermore, the CD3 positivity rate in CD2^+^/CD56^−^ T cells was 99.7% (Appendix A).

Evaluation by co-culture of MoDCs and T cells utilised the following fluorescently labelled anti-human antibodies: CD4-FITC (RPA-T4), 4-1BB-PerCP/Cyanine5.5 (4B4-1) (BioLegend), CD134 (OX40)-PE (ACT35) (BioLegend).

### 2.7. FACS Analysis

Cells for FACS analysis were stained with fluorescently labelled antibodies. Following staining, cells were washed twice with FACS buffer (PBS containing 1% FBS).

Data were collected using FACSymphony A1 (BD Biosciences) and analysed with FCS Express software (FACS FlowJo Software v.10.9.0) to determine the percentage of gated cells positive for each marker and their expression levels. GM-CSF- and IL-4-induced DCs derived from peripheral blood mononuclear cells were defined as CD1c^+^ CD11c^+^ CD14^−^ HLA-DR^+^, respectively.

pDCs were defined as Lineage^−^, HLA-DR^+^, CD123^+^, CD11c^−^, CD1c^−^, CD304^+^ and sorted from PBMCs using FACSMelody (BD Biosciences).

T cells were defined as CD2^+^ and CD56^−^ and sorted. For co-culture analysis, cells were extracted as CD4^+^ and Th1 cells were defined as OX40^+^ and 4-1BB^+^.

### 2.8. Quantitative Polymerase Chain Reaction (qPCR)

Total RNA was extracted from phytochemical-stimulated DCs using the ReliaPrep RNA Miniprep Systems (Promega, Madison, WI, USA). Total RNA was used as a template, and cDNA was synthesized using the PrimeScript RT reagent Kit (Perfect Real Time) (TaKaRa, Shiga, Japan). The procedure was performed according to the product protocol. cDNA samples were added to PCR reactions containing a mixture of specific primers for each pair of *TLR9*, *RIG-I*, *IRF-7*, and *IFN-β*. The reaction mixture was prepared using TB Green Premix Ex Taq II (TaKaRa) according to the protocol, and measurements were performed using the Thermal Cycler Dice^®^ Real Time System III (TaKaRa). Gene expression levels were calculated using the 2^ΔΔCt^ method with the housekeeping gene (*GAPDH*) as a reference. The sequences of the primers used are shown in Appendix A. Relative gene expression under each condition was compared to that of unstimulated peripheral blood mononuclear cell-derived dendritic cells.

### 2.9. Western Blots

After 30 min on ice with intermittent mixing, lysates were clarified at 12,000× *g* for 10 min at 4 °C. Lysates were RIPA lysis buffer (25 mM Tris, 150 mM KCl, 1% NP-40 substitute (Fujifilm wako), 0.5% Sodium Deoxycholate (Fujifilm wako), 5 mM EDTA (Dojindo, Kumamoto, Japan), 0.1% SDS (Fujifilm wako) and 50 mM Sodium Orthovanadate(V) aside at a pH of 7.4), containing proteinase inhibitor cocktail (Sigma-Aldrich). Protein concentration was measured by Pierce™ BCA Protein Assay Kits (Thermo Scientific, Waltham, MA, USA). Proteins from MoDCs cell were analysed by SDS-PAGE. Membranes were blocked with 3%BSA-TBS for 30 min and were incubated with an appropriate primary antibody, 16 h at R.T. Primary antibodies includes anti-RIG-I ((Protein tech, Rosemont, IL, USA), dilution 1:1000). After rinsing, the membranes were incubated with HRP-conjugated secondary antibodies (Rabbit TrueBlot^®^: Anti-Rabbit IgG HRP (ROCKLAND, Philadelphia, PA, USA)) for 1 h. Finally, membranes with incubated with Pierce™ ECL Western Blotting Substrate (Thermo Scientific), chemiluminescent signal was developed and imaged by iBright CL1500 (Thermo Scientific).

### 2.10. Intracellular Flow Cytometry

Phytochemical-stimulated MoDCs were recovered using 1 mM EDTA/PBS, washed, and incubated with Human TruStain FcX™ (BioLegend) for 15 min at R.T. Cells were washed with FACS buffer, fixed with 4% paraformaldehyde for 15 min at R.T., and then permeabilized with 0.01% Triton X-100/PBS for 7 min at R.T. After washing, RIG-1/DDX58 Polyclonal antibody (Protein Tech) was added and incubated for 45 min at 4 °C. After washing, DyLight 488 donkey anti-rabbit IgG (minimal cross-reactivity) (1:300) (BioLegend) was added and incubated for 45 min at 4 °C.

### 2.11. Immunofluorescence Staining

Peripheral blood mononuclear cell-derived dendritic cells were seeded on glass cover slips coated with poly-L-lysine and stimulated with phytochemicals. The cells were washed with 0.5% BSA/PBS, fixed with 4% paraformaldehyde for 15 min, and permeabilised with 0.1% Triton X-100 for 7 min. The cells were washed with 0.5% BSA/PBS and incubated with primary antibodies against RIG-I (Protein Tech) at 37 °C for 1 h. After washing with 0.5% BSA/PBS, DyLight 488 donkey anti-rabbit IgG (minimal cross-reactivity) (1:300) (BioLegend) was added, and the mixture was incubated at 37 °C for 1 h. The cover glass was washed, and the slide was mounted using VECTASHEALD Mounting Medium with DAPI (Vector Laboratories, Newark, CA, USA). Fluorescent Images were acquired on a fluorescence microscope (BZ-X710, KEYENCE, Osaka, Japan) using a Nikon [Plan Fluor x20 NA = 0.45] objective with identical exposure and gain settings across conditions within each donor. Scale bars were generated by the instrument’s optical calibration (objective and camera pixel size) and verified against a stage micrometer (10-µm divisions) before imaging. For each condition, N = [3] independent donors were examined, acquiring 3 fields of view per donor.

### 2.12. Co-Culture Study

Phytochemical-stimulated DCs were collected and grouped according to cell concentration. T cells were fractionated from human peripheral blood using FACSMelody and co-cultured for 1 week. Recombinant human IL-2 (10 U/mL; BioLegend) was added to maintain T cells. Changes in surface markers (OX40/4-1BB) under autologous co-cultured dendritic cell (DC) and T cell conditions were analyzed using FACSymphony A1.

### 2.13. Allogenic Mixed Lymphocyte Reaction (MLR)

Monocyte-derived dendritic cells (MoDCs) were differentiated from human peripheral blood mononuclear cells (PBMCs) and stimulated with a phytochemical for 2 days to induce maturation. CD56^−^ CD2^+^ T cells were allogeneic MLRs isolated using the FACS Melody cell preparative system for a sensitive assessment of DC stimulatory potential. Allogeneic mature monocyte-derived dendritic cells and T cells were co-cultured at a 1:1 ratio (1 × 10^4^ cells/well each) in 96-well round-bottom plates. After 5 days of culture, supernatants were collected, and IFN-γ levels were quantified using an ELISA. Data are presented as the mean ± standard deviation (SD) of three independent experiments.

### 2.14. ELISA

Cytokine concentrations in cell culture supernatants were measured using commercially available ELISA kits. The human IFN-γ kit was purchased from BioLegend. Cytokine production in cell culture supernatants was measured as follows: A 96-well plate (Nunc, Roskilde, Denmark) was coated with capture antibody 16 h at 4 °C. Following washing and blocking, harvested cell culture supernatants were added and incubated at room temperature for 2 h. Avidin-HRP was used for detection. TMB Stop Solution (SeraCare, Milford, MA, USA) was employed as the substrate, and the reaction was stopped by adding 1 M sulfuric acid solution (Fujifilm Wako). Absorbance was measured at 450 nm and 570 nm using SpectraMax^®^ iD3 (MOLECULAR DEVICES, San Jose, CA, USA).

### 2.15. Statistical Analysis

All values are presented as mean ± standard error of the mean (SEM). Data from cytokine production assays and gene expression analyses were analysed using one-way analysis of variance (ANOVA) followed by Dunnett’s test. *N* denotes biological replicates (independent donors or donor pairs). Within each donor, conditions were plated as technical duplicates (averaged to one value per donor to avoid pseudoreplication). All statistical analyses were performed using IBM SPSS Statistics v.29.0.2.0.(20) (IBM, Chicago, IL, USA). A *p* value < 0.05 (*) was considered statistically significant. Data were generated using OriginPro 2025b (OriginLab Corporation, Northampton, MA, USA).

## 3. Results

We confirmed the induction of differentiation from monocytes to MoDCs by cytokine stimulation and performed each analysis using cells under the same conditions (Figure 1).

### 3.1. Phytochemicals Enhance Factors on the MoDCs

Given that DCs harbor antiviral signaling pathways, we assessed whether phytochemicals modulate RIG-I and IFN-β expression. Multiple phytochemicals upregulated the expression of these genes following short-term stimulation, as determined by qPCR (Figure 2A). Since α-mangostin and hesperidin exhibited enhanced gene expression of both *RIG-I* and *IFN-β*, these two compounds were designated as primary targets for measurement. Auraptene, quercetin, and Tangeretin were selected as additional candidate compounds for further analysis. We also profiled the TLR9 pathway (including IRF7) using CpG-ODN 2216 (InvivoGen, San Diego, CA, USA) as a reference ligand. α-Mangostin was active in both the RIG-I and TLR9 pathways, suggesting pathway-specific and time-dependent effects suggesting a pathway-specific and time-dependent effect (Figure 2B,C). These results suggest that although α-Mangostin may be useful in both the RIG-I signaling pathway and the TLR9 signaling pathway, the mechanisms of action may differ.

### 3.2. Changes in RIG-I Expression Were Also Confirmed Within Cells

After confirming the evaluation of gene expression levels using α-Mangostin, we attempted to analyze changes in intracellular expression using various analytical methods. Although changes in intracellular protein expression were observed in samples treated with phytochemicals for 2 days via Western blot analysis under specified exposure times and concentration conditions, no significant differences were noted compared to other plant-derived chemicals (Figure 3A). This suggests that it may be useful to set and analyze the activity response of genes related to antiviral responses within 24 h. Furthermore, in samples treated for 6 h, changes in RIG-I fluorescence intensity were observed using intracellular flow cytometry analysis, demonstrating the usefulness of examining effects over short time periods (Figure 3B).

### 3.3. Imaging Assessment of Intracellular Protein Expression by α-MANGOSTIN

Immunofluorescence staining (RIG-I and DAPI) revealed that α-Mangostin–treated MoDCs exhibited more extensive dendritic process outgrowth compared with unstimulated controls (Figure 4), indicative of maturation-like features in otherwise immature cells.

### 3.4. Co-Culture of Phytochemically Stimulated MoDCs with CD4^+^ T Cells Favors Th1 Cell Differentiation and IFN-γ Production

To evaluate gene expression in dendritic cells (DCs), we co-cultured phytochemical-stimulated mature MoDCs with CD4^+^ T cells at a ratio of 0.5:1. α-Mangostin–treated MoDCs showed an increased proportion of OX40^+^ 4-1BB^+^ T cells, suggesting promotion of Th1 differentiation (Figure 5A). In addition, α-Mangostin significantly enhanced IFN-γ production in co-culture supernatants (Figure 5B), indicating its potential to modulate both innate and adaptive immune responses.

## 4. Discussion

This study demonstrated that α-Mangostin, a representative plant-derived compound, upregulates *RIG-I* (*DDX58*) transcripts in human monocyte-derived dendritic cells (MoDCs), and that this effect is associated with a tendency toward Th1 polarization as well as increased IFN-γ production in co-culture with T cells. These data support the growing body of evidence that plant-derived small molecules can modulate antiviral pathways of innate immunity and shape adaptive immune responses through crosstalk between dendritic cells (DCs) and T cells. Mechanistically, our observations are consistent with the understanding that RIG-I–MAVS signaling in DCs plays a central role in the induction of type I interferons (IFNs), DC activation, and the efficient priming of multifunctional T cell responses [40,41,42].

Plant-derived phytochemicals have been broadly characterized for their antioxidant and anti-inflammatory properties; however, an increasing number of studies have reported their direct effects on pattern recognition receptor (PRR) pathways. For instance, epigallocatechin-3-gallate (EGCG) has been shown to enhance the double-stranded RNA–induced antiviral program in hepatocytes by upregulating TLR3/RIG-I expression and augmenting IFN-λ/ISG responses, suggesting that certain polyphenols can contextually potentiate RLR-mediated antiviral signaling [43]. Conversely, biochemical assays have demonstrated that catechins can inhibit the ATPase activity of RIG-I, highlighting the diverse and context-dependent effects of phytochemicals that vary with assay system, concentration, and cellular environment [44]. Taken together with our findings, these observations suggest that specific plant-derived compounds (and their administration conditions) may skew DCs toward an antiviral state, while not uniformly driving pro-inflammatory outputs.

α-Mangostin is of particular interest, as its activity in primary human monocyte-derived dendritic cells (MoDCs) under viral infection has been reported. Specifically, in MoDCs infected with dengue virus, α-Mangostin (20–25 μM) was shown to reduce viral production while suppressing the transcription of pro-inflammatory cytokines and chemokines [45]. A review article further notes that the immunomodulatory effects of α-Mangostin are mediated through SIRT1 activation, inhibition of the MAPK/NF-κB pathway, and redox activity, pathways that intersect with RLR signaling and IFN production [46]. At first glance, the anti-inflammatory activity reported during infection appears to contradict our observations of increased RIG-I mRNA levels and enhanced IFN-γ production in DC–T cell co-cultures. However, these phenomena can be reconciled as follows: (i) transcriptional priming of RIG-I does not necessarily equate to broad inflammatory cytokine release in the absence of viral PAMPs; (ii) the effects of phytochemicals are stimulus- and time-dependent; and (iii) modulation of upstream sensors may coexist with an overall attenuation of pathological cytokine amplification during infection. Moreover, literature on TLR–RLR cooperation in human DCs supports the notion that moderate enhancement of RLR components can condition subsequent responses to nucleic acid ligands without triggering sustained inflammation [47].

Our CpG ODN2216/pDC experiments expand the scope of analysis beyond MoDCs. CpG-A ODNs, particularly 2216, are well known to induce robust type I IFN secretion from plasmacytoid DCs (pDCs) via TLR9–MyD88–IRF7 signaling [48,49]. The transcriptional changes observed in pDCs following short-term exposure to α-Mangostin suggest the possibility of crosstalk between phytochemical signaling and nucleic acid recognition pathways in this specialized DC subset (Appendix A). Nevertheless, given the limited cell numbers used and the relatively high concentrations applied, further refinement of dose–response relationships and mechanisms of action will be necessary.

In the mixed lymphocyte reaction, MoDCs stimulated with α-Mangostin increased the OX40^+^, 4-1BB^+^ fraction among CD4^+^ T cells and elevated IFN-γ levels in the supernatant. This is consistent with a tendency toward Th1 polarization and/or enhanced effector function. OX40 and 4-1BB are prototypical costimulatory receptors of the TNFR superfamily, whose signaling promotes T cell proliferation, survival, and effector differentiation, and under certain cytokine milieus and antigenic contexts, supports Th1 programming [50,51,52]. However, as OX40 can also support Th2 responses depending on the context, definitive assessment of lineage polarization will require additional markers (e.g., TBX21/T-bet, CXCR3, IL-12 dependency) and evaluation in antigen-specific rather than only allogeneic systems.

The strength of this study lies in the convergent evidence obtained from transcriptional measurements in dendritic cells and functional outputs in DC–T cell co-cultures. Nonetheless, several limitations should be acknowledged. First, MoDCs represent an ex vivo model and do not fully recapitulate primary cDC1/cDC2 subsets in vivo. Second, inter-donor variability in responses to phytochemicals and in baseline RIG-I expression may obscure effect sizes; thus, adequately powered cohorts and paired analyses are warranted. Third, consistent with kinetics of RLR signaling, early (6 h) and intermediate (24 h) windows captured transcript/protein shifts, whereas the 48 h Western blot (late window) showed smaller differences—suggesting transient rather than sustained upregulation. Together, these findings support the possibility that RIG-I signaling dynamics peak within a relatively short timeframe, and that a more detailed time-course analysis (e.g., 2–24 h) may better resolve the discrepancy between transcriptional and protein-level observations. In addition, assessing pathway-proximal readouts—such as phosphorylated TBK1/IRF3, MAVS aggregation by SDD-AGE, and induction of ISGs including MX1, ISG15, and IFIT1—would enable more precise characterization of the signaling activation landscape [5,40,53,54,55]. Fourth, while we observed increased IFN-γ production in co-cultures, the bridging of DC recognition to T cell effector programming is likely mediated, at least in part, by IL-12p70. Measurement of IL-12p70 secretion, as well as blocking experiments with IFNAR1 or IL-12 neutralization, would further strengthen causal inference.

A limitation of this study is the incomplete protein and gene expression profiling. Western blot data could only be obtained for RIG-I, as other target proteins exhibited elevated membrane background and expression levels below the threshold required for accurate densitometric analysis. Similarly, comparative data on IFN-β gene expression in MoDCs subjected to combined phytochemical and CpG stimulation could not be included due to considerable variation during qPCR analysis, which precluded reliable statistical evaluation. Only data for genes and proteins that met quality control criteria are included in this report. Further optimization of experimental protocols and analytical methods will be necessary to comprehensively evaluate these targets in future studies. Future studies should (i) employ defined RIG-I agonists (e.g., 5′-ppp short dsRNA) to confirm pathway specificity and use DDX58 knockdown/CRISPR to test necessity [54]; (ii) dissect cross-talk with TLR9 by combining α-Mangostin with CpG ODNs in both MoDCs and pDCs [46,47,48]; and (iii) expand the compound panel to include phytochemicals with reported DC effects (e.g., resveratrol, EGCG, sulforaphane), recognizing that some agents induce tolerogenic DC phenotypes depending on timing and dose [43,44]. By mapping concentration–time “windows” that prime antiviral readiness (RIG-I/ISGs) without provoking excessive inflammatory cytokines, it may be possible to define nutraceutical strategies that enhance vaccine responsiveness or antiviral defense while minimizing immunopathology.

In broader context, our findings align with the conceptual framework that nutritional cues can tune nucleic acid–sensing pathways in sentinel cells. Given the essential role of RIG-I–MAVS signaling for DC antigen presentation and T-cell priming in vivo, the observation that a dietary-derived compound increases RIG-I transcripts in human DCs and coincides with Th1-leaning functional readouts is biologically and meaningful [40,42,53]. The balance between antiviral preparedness and inflammatory restraint is delicate; α-Mangostin’s dual literature—antiviral/anti-cytokine effects in infected MoDCs and immunomodulatory priming in our non-infected system—suggests that phytochemicals might act as “set-point modulators”, boosting sensor capacity while attenuating excessive downstream amplification [45,46]. Ultimately, delineating structure–activity relationships and defining human-relevant exposure levels will be crucial for moving from in vitro observations to evidence-based nutritional interventions.

## 5. Conclusions

This study demonstrates that the plant-derived component α-Mangostin may enhance the antiviral sensor RIG-I pathway in human dendritic cells and induce Th1-type responses in T cells under co-culture. The dual action of the phytochemicals, being antiviral and anti-inflammatory during infection and promoting immune priming under non-infection, suggests that phytochemicals can function as “set point regulators” of immune responses. In the future, elucidation of optimal conditions for concentration and duration of action, analysis of structure-activity relationships, and verification of effective intake in humans will be essential to realize nutritional interventions that enhance antiviral defenses.

## Figures and Tables

**Figure 1 nutrients-17-03539-f001:**
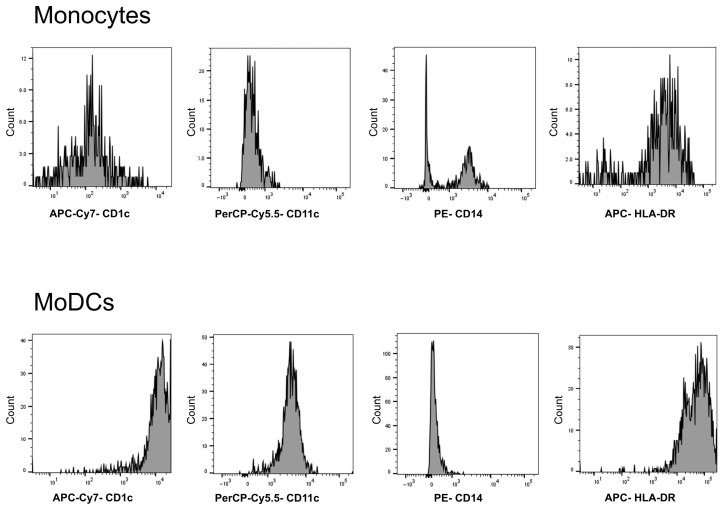
Differentiation of monocytes into monocyte-derived dendritic cells (MoDCs). Representative flow cytometry plots showing gating of peripheral blood mononuclear cells (PBMCs) and identification of MoDCs (CD1c^+^, CD11c^+^, CD14^−^, and HLA-DR^+^). They were cultured for one week with the addition of GM-CSF (5 ng/mL) and IL-4 (10 ng/mL). PBMCs prior to culture and MoDCs post-culture were analyzed using FACS. The horizontal axis is shown on a logarithmic scale ranging from 10^−3^ to 10^5^. APC-Cy7-CD1c; Allophycocyanin-Cyanine7 conjugate anti-CD1c antibody. PerCP-Cy5.5-CD11c, Peridinin chlorophyll protein-Cyanin5.5 conjugate anti-CD11c antibody; PE-CD14, Phycoerythrin conjugate anti-CD14 antibody; APC-HLA-DR, Allophycocyanin conjugate anti-human leukocyte antigen-DR antibody; MoDCs, Monocyte-Derived Dendritic Cells.

**Figure 2 nutrients-17-03539-f002:**
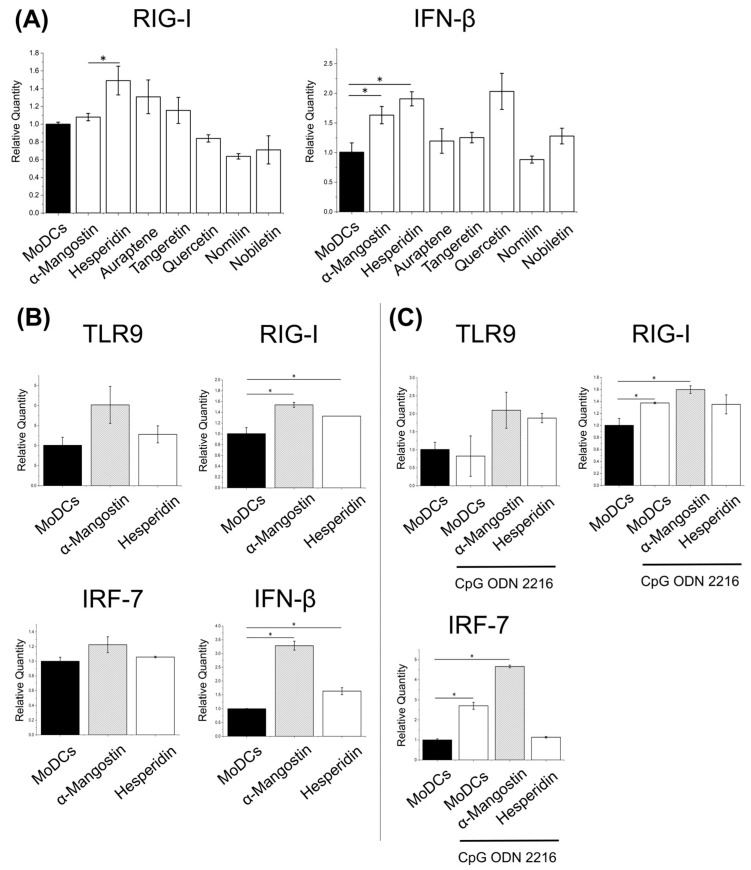
Gene expression levels upon phytochemical stimulation. MoDCs were differentiated from PBMCs via cytokine stimulation. (**A**). Phytochemicals were added to MoDCs seeded at 2 × 10^5^ cells/mL at 5 μM each and incubated for 6 h. (**B**). Phytochemicals were added at 5 µM each to MoDCs seeded at 2 × 10^5^ cells/mL and incubated for 1 day. (**C**). MoDCs treated for 1 day under the same conditions as (**B**) in the presence of phytochemicals were stimulated with an additional 1 μM of CpG-ODN 2216 and cultured for an additional 1 day. Gene expression levels of *TLR9*, *RIG-I*, *IRF-7*, and *IFN-β* were measured by qPCR. ***: *p* < 0.05 compared to MoDCs alone and each sample. Data represent the mean ± standard deviation (SD) from three independent measurements. TLR9, Toll-like receptor 9; RIG-1, Retinoic acid-inducible gene-I; IRF-7, Interferon Regulatory Factor 7; IFN-B, Interferon-beta; MoDCs, Monocyte-Derived Dendritic Cells; CpG ODN 2216, Cytosine-phosphate-guanine oligodeoxynucleotide 2216.

**Figure 3 nutrients-17-03539-f003:**
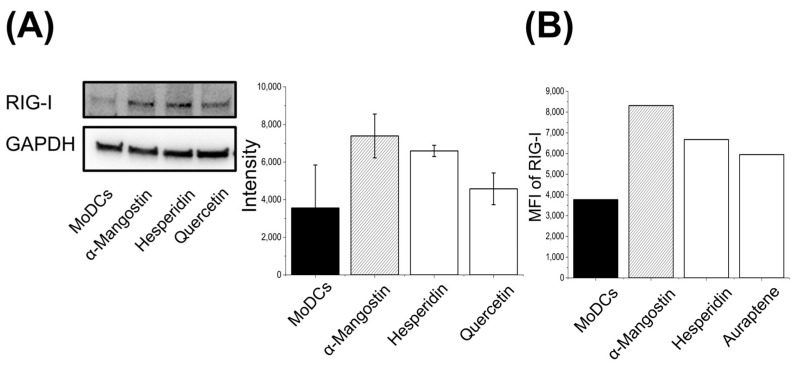
Molecular mechanism of RIG-I activation in MoDCs. MoDCs were differentiated from PBMCs and seeded at 3 × 10^5^ cells/mL. (**A**) Phytochemicals 5 µM for 48 h; lysates analysed by Western blot (RIG-I). Data represent the mean ± standard deviation (SD) from three independent measurements. (**B**) Phytochemicals 5 µM for 6 h; intracellular flow cytometry for RIG-I. RIG-1, Retinoic acid-inducible gene-I; GAPDH, Glyceraldehyde-3-Phosphate Dehydrogenase; MoDCs, Monocyte-Derived Dendritic Cells; MFI, Mean fluorescence intensity.

**Figure 4 nutrients-17-03539-f004:**
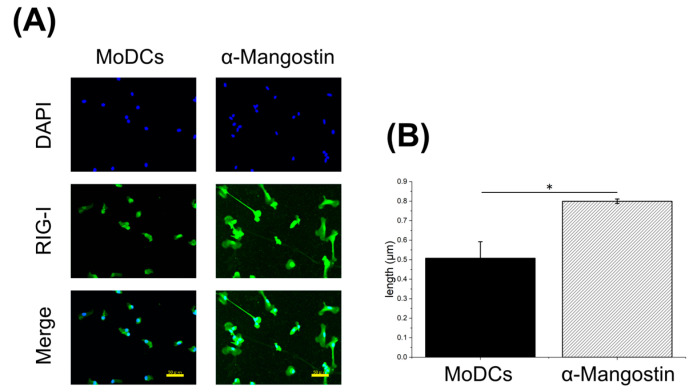
Immunofluorescence staining of MoDCs. MoDCs were differentiated from PBMCs obtained from peripheral blood. MoDCs seeded at 3 × 10^5^ cells/mL were incubated with 5 μM α-Mangostin for 1 day. (**A**) DAPI and RIG-I staining in cells from the unstimulated group and the α-Mangostin group (measured at 3 fields per group. Scale bar: 50 μm). (**B**) Quantification of dendrite length in each cell). Fluorescent images were acquired using a fluorescence microscope equipped with a Nikon Plan Fluor 20×/0.45 NA objective. Identical exposure and gain settings were applied across all conditions within each donor. Dendritic length was quantified by averaging three fields of view per donor, and data are presented as means ± standard error of the mean (SEM) at the donor level. A two-tailed unpaired *t*-test was performed to compare vehicle- and α-mangostin–treated groups, with statistical significance defined as *: *p* < 0.05. RIG-1, Retinoic acid-inducible gene-I; MoDCs, Monocyte-Derived Dendritic Cells; DAPI, 4′,6-diamidino-2-phenylindole.

**Figure 5 nutrients-17-03539-f005:**
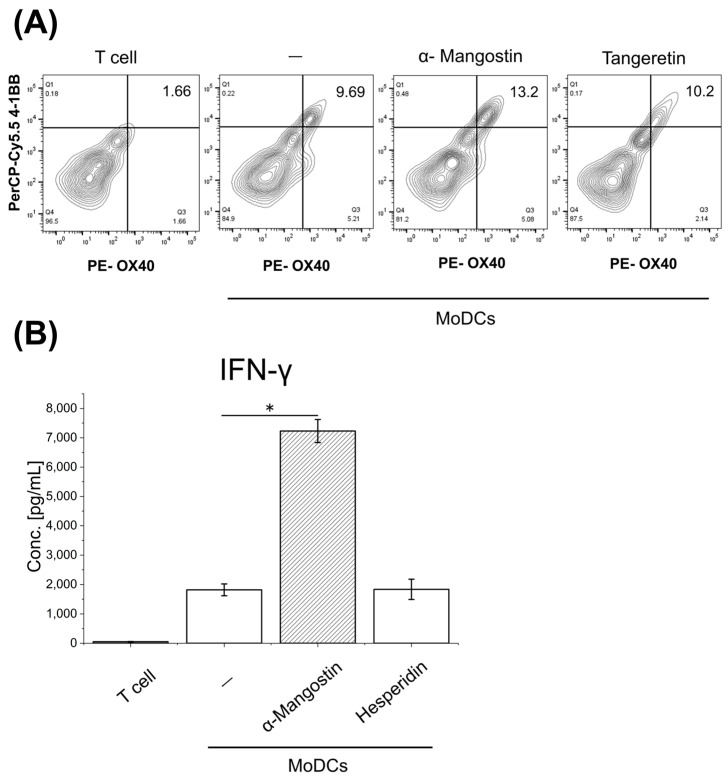
Effects of MoDCs and T Cells on T Cell Differentiation in Mixed Culture. PBMCs were used to generate MoDCs and isolate T cells. MoDCs were treated with 5 μM of each phytochemical for 2 days. (**A**) Phytochemical-stimulated MoDCs were co-cultured with T cells at a 0.5:1 ratio for 1 week. T cell differentiation in co-culture was analyzed by FACS. CD4^+^ cells were gated, and OX40^+^ (PE) and 4-1BB^+^ (PerCP/Cy5.5) were used as markers of Th1 differentiation. (**B**) Lymphocytes from different donors were used. The high sensitivity of DC stimulation ability by homologous MLR was evaluated. Two volunteer donors (male) MoDCs and T cells were co-cultured at a 1:1 ratio for 5 days, and supernatants were analyzed by ELISA. Bars represent mean ± SEM at the donor-pair level (duplicate wells averaged). One-way ANOVA with Dunnett’s test was performed vs. Vehicle-conditioned MoDCs + T cells. Significance is indicated as: * *p* < 0.05. PerCP-Cy5.5 4-1BB, Peridinin chlorophyll protein-Cyanin5.5 conjugate anti-4-1BB antibody; PE-OX40, Phycoerythrin conjugate anti-OX40 antibody; MoDCs, Monocyte-Derived Dendritic Cells; Conc., concentration; IFN-γ, Interferon-gamma.

## Data Availability

The original contributions presented in this study are included in the article. Further inquiries can be directed to the corresponding author.

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
