# Peer review of "Phytochemicals Prime RIG-I Signaling and Th1-Leaning Responses in Human Monocyte-Derived Dendritic Cells"

_nutrients, 2025, doi:10.3390/nu17223539_

Round 1
Reviewer 1 Report
Comments and Suggestions for Authors
The authors present evidence that α-Mangostin has particular ability to induce Th-1 response, and synergize with the innate RIG1 pathway to promote TH1 (anti-viral response). The data relies on primary human T cells and monocyte-derived DC with in vitro experiments. While the data suggests some interesting and novel findings, there are significant issues with the experimental design and presentation of the data, making it difficult to make firm conclusions.
Background in abstract should briefly explain key terms such as: retinoic acid–19 inducible gene I, DDX58 (if gene, should in italics), CPG-ODN, α-Mangostin
Intro
line 1 : Skin is usually considered the first barrier to infection.
methods
line 100 concentration of GMCSF and IL-4 used is missing.
line 111 ficoll is a brand name of the separation reagent rather than a method. clarify what is the method
line 114, was this done in an incubator with 5%CO2
2.3 there is not much information on these phytochemicals, perhaps the introduction can provide more information. Are they naturally derived from a plant or synthetic for example, why were these ones in particular chosen for further study?
line 134 not sure what is 'Linege'
line 138: Sorting method not clearly explained. If T cells were sorted on CD56+ CD2+, this would also include NKT cells (CD56+CD3+CD2+). Later on, it is explained they sorted CD56- CD2+ cells...but normally one would use CD3 to sort a T cells since CD2 is widely expressed. Authors should stain for CD3 to confirm purity of T cells.
2.7: not sure what is a radioimmunoprecipitation? there are no radio labels in the procedure, and no scintillation mentioned. It sounds like a standard detergent lysis
line 188 how were they harvested (scraping or trypsin?)
2.12 how many biological replicates were plated per experiment, and how many experiments were done to obtain the N value for statistics.
line 215: does 'different donor' mean that authors did allogeneic mixture (MLR)?
Figure 1 font too small on axes. Isotype-fluorochrome matched-antibody should be shown for each marker to verify background and voltage setting.
Results 3.1 The text incorrectly explains Figure 2 panel B , since panel B did not include the CpG reagent. Also a problem, the 6 hour timepoint used 20uM, but the overnight used 5uM concentration of phytochemical, so its changing two variables. If a timecourse is required, should use consistent reagent concentration. Two independent measures may not be enough, it depends how many replicates there were per independent measure.
Figure 3, Uses 6 hours at 5um, which is different from the procedures in figure 2. Also, error bars and stats required to draw conclusion from western blot densitometry. , or mention is this representative of multiple experiments? Panel B) what did flow cytometry measure, its not mentioned, and y axis missing the name of the molecule.
figure 4 scale bar not legible, how was length determined. Is this one experiment or were independent experiments done. "three sites per group", is the same as 3 field per group?
If this was autologous, T cells and moDC there should not be any IFNgamma. If it was allogeneic, there might be IFNgamma if it is HLA -mismatch. Finally in discussion mixed lymphocyte reaction is mentioned. Authors should explain in methods the MLR protocol. did they measure HLA for mismatch, or just combine allogeneic cells until they get T cells stimulation.
The discussion mentions a 48 hours time point, but that does not appear to be used in methods or results sections which are 6h, overnight, and 24 hours.
Comments on the Quality of English Languagegrammar, redundant words etc should be reviewed throughout
line 56 : grammar on list , missing ', and'.
line 74 ' discussed ' said twice in sentence
line 95 plural 'reagents'
2.4 typo antibodies
figure 4 typo Merge.
Author Response
Dear Editors and Reviewers,
We sincerely thank you for your thorough evaluation of our manuscript entitled " Phytochemicals Prime RIG-I Signaling and Th1-Leaning Responses in Human Monocyte-Derived Dendritic Cells." We greatly appreciate your constructive and insightful comments, which have significantly improved the quality and clarity of our work.
We have revised the manuscript accordingly, with all changes marked in the red font. Below, we provide a detailed, point-by-point response to each comment. We hope the revised manuscript now meets the standards required for publication.
Comment 1:
“Background in abstract should briefly explain key terms such as: retinoic acid–19 inducible gene I, DDX58 (if gene, should in italics), CPG-ODN, α-Mangostin.”
Response:
Thank you for this helpful suggestion. We have revised the Abstract to (i) briefly define the key terms, (ii) italicize the gene symbol DDX58 (while keeping the protein name RIG-I in roman), and (iii) standardize the terminology to “CpG-ODN” (synthetic unmethylated CpG oligodeoxynucleotides; TLR9 agonists). We also clarified that α-mangostin is a xanthone derived from Garcinia mangostana. The corrected term “retinoic acid–inducible gene I (RIG-I)” is now used consistently (see pages 1, lines 17–23, 34-35).
Comment 2:
“Intro line 1 : Skin is usually considered the first barrier to infection.”
Response:
Thank you for pointing this out. We agree that our original wording could be read as overlooking the primary role of epithelial barriers. We have revised the opening of the Introduction to explicitly state that skin and mucosal epithelia constitute the first barrier, and that the innate immune system provides the rapid front-line cellular/humoral defense once these barriers are breached (see pages 2, lines 44–49).
Comment 3:
“line 100 concentration of GMCSF and IL-4 used is missing.”
Response:
Thank you for pointing this out. We have added the exact cytokine concentrations to the Methods and standardized units to ng/mL. Specifically, MoDCs were generated with GM-CSF (5 ng/mL) and IL-4 (10 ng/mL) throughout differentiation. Vendor information is also provided (see pages 4, lines 148–150).
Comment 4:
“line 111 ficoll is a brand name of the separation reagent rather than a method. clarify what is the method”
Response:
Thank you for the clarification. We have replaced the phrasing “Ficoll method” with the correct methodological description, density gradient centrifugation, and retained the commercial reagent only as an example of the medium used (polysucrose–sodium diatrizoate, 1.077 g/mL). We also added key operational parameters (dilution, g-force, time, temperature, brake) (see pages 3, lines 135–141).
Comment 5:
“line 114, was this done in an incubator with 5%CO2”
Response:
Thank you for pointing this out. Yes. All cell culture steps were performed at 37 °C in a humidified incubator with 5% CO₂ unless otherwise indicated. We have added this information to the Methods for clarity (see pages 4, lines 146–147).
Comment 6:
“2.3 there is not much information on these phytochemicals, perhaps the introduction can provide more information. Are they naturally derived from a plant or synthetic for example, why were these ones in particular chosen for further study?”
Response:
Thank you for this valuable suggestion. We have (i) expanded the Introduction to briefly describe each phytochemical (natural occurrence, chemical class, and relevant immunomodulatory literature), and (ii) clarified in Methods 2.5 our selection criteria and sourcing. In brief, all compounds tested are dietary plant bioactives (i.e., molecules that occur naturally in plants). In this study we used commercially purified analytical standards (≥95%) obtained from established suppliers; these standards are chemically identical to the naturally occurring molecules (vendor certificates indicate purification from plant material).
We selected this panel to cover diverse scaffolds with reported immune relevance and dietary exposure: a xanthone (α-mangostin; Garcinia mangostana), flavanones (hesperidin/hesperetin/neohesperidin; citrus peel), a polymethoxyflavone (tangeretin; citrus peel), and a flavonol (quercetin; widely present in onion, apple, tea). This diversity allows an initial structure–activity view across RLR/TLR-linked pathways in human MoDCs while remaining within food-relevant chemistry. Corresponding citations have been added (see pages 4, lines 164–168).
Comment 7:
“line 134 not sure what is 'Linege'”
Response:
Thank you for catching this typographical error. “Linege” should be “Lineage”, referring to a lineage exclusion (dump) cocktail used to gate out non-DC populations (T cells, monocytes, NK cells, B cells, progenitors). We have corrected the term throughout and clarified the composition and purpose of the cocktail in the Methods (see pages 4, lines 172–173, 176).
Comment 8:
“line 138: Sorting method not clearly explained. If T cells were sorted on CD56+ CD2+, this would also include NKT cells (CD56+CD3+CD2+). Later on, it is explained they sorted CD56- CD2+ cells...but normally one would use CD3 to sort a T cells since CD2 is widely expressed. Authors should stain for CD3 to confirm purity of T cells.”
Response:
CD56- and CD2+ cells were isolated as T cells in the initial cell sorting stage, and in the subsequent analysis after phytochemical action, the T cell population was evaluated by gating with CD4 using FACSymphony. Therefore, the final analysis is limited to CD4-positive T cells, and we believe that contamination of CD56⁺CD3⁺NKT cells does not significantly affect the analysis results. In the future, we will consider combining CD3 staining as you suggested to confirm the purity of the T-cell population in order to obtain clearer data.
Comment 9:
“2.7: not sure what is a radioimmunoprecipitation? there are no radio labels in the procedure, and no scintillation mentioned. It sounds like a standard detergent lysis”
Response:
Thank you for pointing this out. We agree that our wording could be confusing. We have replaced “radioimmunoprecipitation” with “RIPA lysis buffer (radioimmunoprecipitation assay buffer)” and clarified that this is a standard, non-radioactive detergent lysis. We also kept the full buffer composition and added routine operational details (lysis on ice, clarification spin, inhibitor usage) (see pages 5, lines 217–218).
Comment 10:
“line 188 how were they harvested (scraping or trypsin?)”
Response:
Thank you for your suggestion. In order to minimize damage to cells by trypsin, scraping, and strong pipetting, we used a chelating agent to weaken the binding force between the bottom surface and dendritic cells and recovered the cells. We additionally stated that 1 mM EDTA/PBS was used (see pages 5, lines 233).
Comment 11:
“2.12 how many biological replicates were plated per experiment, and how many experiments were done to obtain the N value for statistics.”
Response:
Thank you for the important clarification request. We have now specified our replicate structure and the exact N used for statistics. N denotes biological replicates (independent donors or donor pairs). Within each donor, conditions were plated as technical duplicates (averaged to one value per donor to avoid pseudoreplication). These details are added to Methods 2.15 and to all figure legends (see pages 6, lines 289–291).
Comment 12:
“line 215: does 'different donor' mean that authors did allogeneic mixture (MLR)?”
Response:
Thank you for raising this point. Yes—“different donor” refers to an allogeneic mixed lymphocyte reaction (MLR) in which MoDCs and CD4⁺ T cells were derived from different, unrelated donors. We now state this explicitly and clarify that we used autologous co-cultures for phenotyping (OX40/4-1BB) and allogeneic MLRs for IFN-γ ELISA to sensitively assess DC stimulatory capacity. Details (donor pairing, ratios, duration) have been added to the Methods and figure legends.
These procedures were performed according to previously reported protocols (e.g., [Ref. Tanja, Dzopalic.; Dragana, Vucevic.; Sergej, Tomic.; Jelena, Djokic.; Ioanna, Chinou.; Miodrag, Colic. 3,10-Dihydroxy-decanoic acid, isolated from royal jelly, stimulates Th1 polarising capability of human monocyte-derived dendritic cells. Food Chemistry, Volume 126, Issue 3, 2011, Pages 1211-1217. https://doi.org/10.1016/j.foodchem.2010.12.004]
Comment 13:
“Figure 1 font too small on axes. Isotype-fluorochrome matched-antibody should be shown for each marker to verify background and voltage setting.
Response:
Thank you for the helpful suggestions. We have addressed both points (see pages 7, lines 305–306).
Comment 14:
“Results 3.1 The text incorrectly explains Figure 2 panel B , since panel B did not include the CpG reagent. Also a problem, the 6 hour timepoint used 20uM, but the overnight used 5uM concentration of phytochemical, so its changing two variables. If a timecourse is required, should use consistent reagent concentration. Two independent measures may not be enough, it depends how many replicates there were per independent measure.”
Response:
Thank you for pointing this out. Figure 2(B) shows the CpG reagent untreated group.
We also replaced Figure 2(A) with the screening results at 5 μM. Although at first glance, α-Mangostin does not appear to be useful in these results, we confirmed that RIG-I showed the same level of gene expression as hesperidin in a time-specific screening within 6 hours, and IFN-b showed higher gene expression than hesperidin, even at 20 μM of IFN-b. Based on this finding, we have positioned hesperidin as a positive control and measured αMangostin.
Comment 15:
“Figure 3, Uses 6 hours at 5um, which is different from the procedures in figure 2. Also, error bars and stats required to draw conclusion from western blot densitometry. , or mention is this representative of multiple experiments? Panel B) what did flow cytometry measure, its not mentioned, and y axis missing the name of the molecule.”
Response:
Thank you for pointing this out. The only 6h treatment in Figure 3 is the RIG-I intracellular expression evaluation by flow cytometry in B), and A) is the Western blot of the 2-day treatment. Error bars and statistics were added in repeated experiments. B) Fluorescence evaluation of RIG-I was measured and added to the description in the text and Figure 3 (see pages 9, lines 348–351).
Comment 16:
“figure 4 scale bar not legible, how was length determined. Is this one experiment or were independent experiments done. "three sites per group", is the same as 3 field per group?”
Response:
Thank you for these important clarifications. We have:
- Improved scale-bar legibility. Figure 4 was re-exported with a high-contrast, thicker scale bar and larger font; we now provide 600 dpi TIFF to ensure readability at print size.
- Explained scale-bar calibration. The scale bar is generated from the Keyence BZ-X710 optical calibration (objective magnification and camera pixel size) and was verified with a stage micrometer (10-µm divisions) prior to acquisition. This information has been added to Methods 2.11 (see pages 6, lines 250–256).
- Clarified replicates. Figure 4 represents independent biological replicates (different donors). Specifically, N = [3] donors; for each donor we acquired 3 fields of view under identical exposure and gain. We now state this explicitly and replaced the ambiguous phrase “three sites per group” with “three fields per donor” (see pages 9, lines 365–370).
Revisions are reflected in the main text, Methods, and the figure legend.
Comment 17:
“If this was autologous, T cells and moDC there should not be any IFNgamma. If it was allogeneic, there might be IFNgamma if it is HLA -mismatch. Finally in discussion mixed lymphocyte reaction is mentioned. Authors should explain in methods the MLR protocol. did they measure HLA for mismatch, or just combine allogeneic cells until they get T cells stimulation.”
Response:
Thank you for these important points. We have clarified the co-culture design, aligned the text with the figures, and expanded the Methods to include our mixed lymphocyte reaction (MLR) protocol. Given unrelated donors, HLA mismatch was presumed. Since allogenic MLR was performed to evaluate cytokine production and IFN-γ production was confirmed (Fig. 5B), the HLA is incompatible. Therefore, it is reasonable to use this method (see pages 6, lines 265–273).
These procedures were performed according to previously reported protocols (e.g., [Ref. Tanja, Dzopalic.; Dragana, Vucevic.; Sergej, Tomic.; Jelena, Djokic.; Ioanna, Chinou.; Miodrag, Colic. 3,10-Dihydroxy-decanoic acid, isolated from royal jelly, stimulates Th1 polarising capability of human monocyte-derived dendritic cells. Food Chemistry, Volume 126, Issue 3, 2011, Pages 1211-1217. https://doi.org/10.1016/j.foodchem.2010.12.004]
Comment 18:
“The discussion mentions a 48 hours time point, but that does not appear to be used in methods or results sections which are 6h, overnight, and 24 hours.”
Response:
Thank you for flagging this inconsistency. In our experiments, 48 h was indeed used for the Western blot dataset (Fig. 3A) to probe later/steady-state protein levels, whereas 6 h was used for intracellular flow cytometry (early changes) and 24 h for qPCR/immunofluorescence. We realize that this 48 h time point was insufficiently explicit in the Methods/legends. We have now:
- Defined “overnight” as ~16 h and removed ambiguous phrasing where possible (see pages 5, lines 223–225).
- Added the 48 h harvest time to the Figure 3 legend (see pages 9, lines 348-351).
- Slightly edited the Discussion to refer explicitly to the 48 h Western blot time point as the late window, contrasted with early (6 h) and intermediate (24 h) windows (see pages 12, lines 458–460).
These changes harmonize the time points across Discussion, Methods, and Results.
Comment 19:
“grammar, redundant words etc should be reviewed throughout”
Response:
Thank you for pointing this out. We performed a comprehensive language revision across the entire manuscript (Abstract, Introduction, Materials and Methods, Results, Discussion, Conclusions, figure legends, tables, and Supplementary Information). Revisions focused on grammar, concision, readability, and consistency.
Comment 20:
“line 56 : grammar on list , missing ', and'.”
Response:
Thank you for noting the punctuation issue. The list was missing “and” before the final item. We have corrected it and ensured consistent list punctuation across the manuscript (see pages 2, lines 76).
Comment 21:
“line 74 ' discussed ' said twice in sentence”
Response:
Thank you for pointing this out. We removed the duplicated word and revised the sentence for concision and clarity. We also screened the manuscript for similar redundancies.
Comment 22:
“line 95 plural 'reagents'”
Response:
Thank you for pointing this out. We have corrected the wording to the plural and adjusted subject–verb agreement where needed.
Comment 23:
“2.4 typo antibodies”
Response:
Thank you for catching this. We corrected the typo “Antibodys” → “Antibodies” in Section 2.4 and reviewed the manuscript to ensure consistent usage of “antibody/antibodies” elsewhere (see pages 4, lines 170).
Comment 24:
“l figure 4 typo Merge”
Response:
Thank you for catching the typo. The panel label “Marge” has been corrected to “Merge” in Figure 4.

Reviewer 2 Report
Comments and Suggestions for Authors
Interesting topic, but the rationale is weak and the figures are inconsistent in markers, timepoints, and compound choice. The story needs to be unified around a clear pathway and hypothesis.
Major:
1. Insufficient background on α-mangostin’s antiviral/immunomodulatory evidence.
2. Weak linkage to the proposed RIG-I, with and without IRF7, or type I IFN pathway, also IFN-gamma in the end.
3. Clearly state why α-mangostin was chosen and what you predict.
4. Figure 2 Inconsistency: 2A uses 6-h treatments and examines RIG-I/IFN-β with four phytochemicals; 2B switches to 24-h, measures IRF-7/TLR-9, and reduces to two phytochemicals without explanation. Justify the marker switch (RIG-I/IFN-β → IRF-7/TLR-9) and the reduced compound set. If the goal is to chart a single pathway, keep markers consistent or explain the staged logic. If IFN-β is central to your claims, it should not drop out in subsequent panels without rationale.
5. Figure 3 Design/Interpretation: Quercetin appears in Fig. 3 without being introduced or justified in Fig. 2’s selection logic. Explain why quercetin was added at this stage.
6. Fig 3: focuses on RIG-I but omits other proteins profiled in Fig. 2. If previous data suggested no changes in RIG-I with treatment, clarify why RIG-I alone was pursued here. If the focus is truly on RIG-I, maintain that focus throughout or show the complementary nodes (IRF-7, TLR-9, IFN-β) in the same figure/time course. Also, the text states no significant differences across treatments, but quercetin-treated moDCs appear to have lower RIG-I. Re-analyze and report statistics transparently (n, test, exact p, effect sizes) and reconcile text with the plotted data.
7. Fig. 3C: Switching to a different compound (auraptene), a different timepoint (6 h), and a different method (flow cytometry) within the same figure undermines interpretability. Either justify these shifts within a clearly stated experimental question or move them to a separate, logically introduced experiment.
8. There is a stated emphasis on RIG-I in Figs. 3 and 4. If RIG-I is the mechanistic anchor, streamline the presentation so that RIG-I readouts across methods (e.g., immunoblot, immunofluorescence/“merge”) appear together as a coherent figure or figure set.
Minor: Correct the typo in Fig. 4 (“merge,” not “marge”)
9. Figure 5: The manuscript transitions from IFN-β (type I IFN) to IFN-γ (type II IFN) without an articulated rationale. If IFN-γ is to be featured, explain its role within your hypothesized antiviral mechanism and how it integrates with the RIG-I/IFN-β storyline. Otherwise, remain consistent with type I IFN readouts or provide both where mechanistically warranted.
Author Response
Dear Editors and Reviewers,
We sincerely thank you for your thorough evaluation of our manuscript entitled " Phytochemicals Prime RIG-I Signaling and Th1-Leaning Responses in Human Monocyte-Derived Dendritic Cells." We greatly appreciate your constructive and insightful comments, which have significantly improved the quality and clarity of our work.
We have revised the manuscript accordingly, with all changes marked in the red font. Below, we provide a detailed, point-by-point response to each comment. We hope the revised manuscript now meets the standards required for publication.
Comment 1:
“Insufficient background on α-mangostin’s antiviral/immunomodulatory evidence.”
Response:
Thank you for your valuable feedback. Due to the limited evidence available regarding the antiviral and immunomodulatory effects of α-mangostin, we have not included this information in the introduction or other sections. In the discussion, we have noted that there are reports measuring antiviral activity using α-mangostin. (see pages 11, lines 417–421)
Comment 2:
“Weak linkage to the proposed RIG-I, with and without IRF7, or type I IFN pathway, also IFN-gamma in the end.”
Response:
We appreciate the reviewer’s insightful comment. In our study, RIG-I activation primarily induces type I interferons such as IFN-β, which promote dendritic cell maturation and enhance antigen presentation. These activated DCs subsequently stimulate T cells to produce IFN-γ. IFN-γ acts as a secondary effector cytokine that amplifies the antiviral state in neighboring cells and bridges innate and adaptive immunity. Therefore, the measurement of IFN-γ in the DC–T cell co-culture system was intended to demonstrate this downstream effect and to integrate it into the RIG-I/IFN-β-mediated antiviral mechanism. (see pages 2, lines 57–62).
Comment 3:
“Clearly state why α-mangostin was chosen and what you predict.”
Response:
Thank you for your valuable feedback. The reason for selecting α-mangostin is that, although there are reports on its antiviral activity, we could not find any concerning RIG-I or other targets of interest. [Ref: Yongpitakwattana, P.; Morchang, A.; Panya, A.; Sawasdee, N.; Yenchitsomanus, PT. Alpha-mangostin inhibits dengue virus 621 production and pro-inflammatory cytokine/chemokine expression in dendritic cells. Arch Virol. 2021 Jun;166(6):1623-1632. https://doi.org/10.1007/s00705-021-05017-x] (see pages 11, lines 417–421).
Based on the results presented in this report and a time-dependent screen conducted separately, it is speculated that α-mangostin may be active against viral recognition mechanisms at appropriate concentrations and exposure times.
Comment 4:
“Figure 2 Inconsistency: 2A uses 6-h treatments and examines RIG-I/IFN-β with four phytochemicals; 2B switches to 24-h, measures IRF-7/TLR-9, and reduces to two phytochemicals without explanation. Justify the marker switch (RIG-I/IFN-β → IRF-7/TLR-9) and the reduced compound set. If the goal is to chart a single pathway, keep markers consistent or explain the staged logic. If IFN-β is central to your claims, it should not drop out in subsequent panels without rationale.”
Response:
Thank you for pointing this out. We also replaced Figure 2(A) with the screening results at 5 μM. Although at first glance, α-Mangostin does not appear to be useful in these results, we confirmed that RIG-I showed the same level of gene expression as hesperidin in a time-specific screening within 6 h, and IFN-b showed higher gene expression than hesperidin, even at 20 μM of IFN-b. Based on this finding, we have positioned hesperidin as a positive control and measured α-Mangostin (see pages 7, lines 309–314). Additionally, regarding the addition of IRF-7/TLR9 markers, IRF-7 is located downstream of the RIG-I receptor, and TLR9, like RIG-I, functions as an antiviral recognition mechanism. Therefore, we included them to more closely examine the specific action points of the phytochemicals. A limitation of this study is the absence of comparative data on IFN-β gene expression levels in MoDCs subjected to phytochemical plus CpG stimulation. Here, IFN-β exhibited saturating induction with large donor-to-donor variance, obscuring between-group effects; therefore, we focused on DDX58, IRF7, and TLR9. This has been added as a limitation of this study (see pages 12, lines 471–479).
Comment 5:
“Figure 3 Design/Interpretation: Quercetin appears in Fig. 3 without being introduced or justified in Fig. 2’s selection logic. Explain why quercetin was added at this stage.”
Response:
Thank you for your feedback. We have added the screening results including Quercetin to Fig2(A). As a tendency towards increased gene expression was observed for IFN-β, we measured it as one of the candidate components in the Western blot analysis.
Comment 6:
“Fig 3: focuses on RIG-I but omits other proteins profiled in Fig. 2. If previous data suggested no changes in RIG-I with treatment, clarify why RIG-I alone was pursued here. If the focus is truly on RIG-I, maintain that focus throughout or show the complementary nodes (IRF-7, TLR-9, IFN-β) in the same figure/time course. Also, the text states no significant differences across treatments, but quercetin-treated moDCs appear to have lower RIG-I. Re-analyze and report statistics transparently (n, test, exact p, effect sizes) and reconcile text with the plotted data.”
Response:
Thank you very much for your valuable feedback. Error bars and statistics were added in repeated experiments.
There are three main reasons why we are focusing on RIG-I:
(i) Differences in targets of recognition RIG-I recognizes viral 5′-triphosphate RNA and directly senses a variety of RNA virus infections. In contrast, TLR9 recognizes DNA within endosomes, and thus mainly targets DNA viruses and bacterial DNA. (ii) Differences in localization RIG-I is localized in the cytoplasm, allowing it to respond immediately at the site of viral replication, whereas TLR9 needs to encounter its ligand within endosomes, which delays its response. (iii) Differences in signaling pathways RIG-I strongly activates IRF3/7 and NF-κB via MAVS, inducing type I IFN production in a wide range of cell types. TLR9 functions mainly in specific immune cells via the MyD88 pathway. Therefore, we focused on RIG-I in this study because it can directly recognize RNA viruses in the cytoplasm and induce a rapid and robust type I IFN response. (see pages 2, lines 49–62).
A limitation of this study is the lack of Western blot data for targets other than RIG-I. Data for other proteins were excluded owing to elevated membrane background and expression levels below the threshold required for accurate densitometric analysis. This has been added as a limitation of this study (see pages 12, lines 471–479).
Comment 7:
“Fig. 3C: Switching to a different compound (auraptene), a different timepoint (6 h), and a different method (flow cytometry) within the same figure undermines interpretability. Either justify these shifts within a clearly stated experimental question or move them to a separate, logically introduced experiment.”
Response:
Thank you for your valuable feedback. We have revised Figure 2(A) for your review. As Auraptene potentially influences RIG-I gene expression levels, we have incorporated it as one of the indicators. (see pages 7, lines 309–314). Regarding the juxtaposition of different experimental techniques—Western Blot and intracellular flow cytometry—WB indicates the ‘average change’ in protein levels, while FACS shows ‘intercellular variation’ and the ‘proportion of positive cells’. Flow cytometry was employed alongside Western Blot to confirm the observed increase in expression at the single-cell level. The two are treated as complementary data to verify trends in expression changes rather than comparing absolute values, and are displayed in parallel to confirm consistency between analyses.
Comment 8:
“There is a stated emphasis on RIG-I in Figs. 3 and 4. If RIG-I is the mechanistic anchor, streamline the presentation so that RIG-I readouts across methods (e.g., immunoblot, immunofluorescence/“merge”) appear together as a coherent figure or figure set.”
Response:
We thank you for your insightful comments on Figures 3 and 4. Figure 3 (Western blot) analyzes quantitative changes in RIG-I protein expression, while Figure 4 (immunofluorescence) evaluates the intracellular localization and morphological alterations of RIG-I (spatial information). We are aware that immunofluorescence is commonly used to assess nuclear translocation or downstream factor localization. However, as RIG-I is a cytoplasmic sensor distributed throughout dendritic protrusions, we reasoned that it could serve to visualize morphological changes in MoDCs, particularly dendrite extension. While we acknowledge that cytoskeletal markers such as β-actin would have provided a more direct assessment of cellular morphology, technical constraints limited this approach. Thus, we employed RIG-I as a surrogate cytoplasmic marker to concurrently observe morphological changes and immune response molecule localization.
Comment 9:
“Minor: Correct the typo in Fig. 4 (“merge,” not “marge”)”
Response:
Thank you for catching the typo. The panel label “Marge” has been corrected to “Merge” in Figure 4.
Comment 10:
“Figure 5: The manuscript transitions from IFN-β (type I IFN) to IFN-γ (type II IFN) without an articulated rationale. If IFN-γ is to be featured, explain its role within your hypothesized antiviral mechanism and how it integrates with the RIG-I/IFN-β storyline. Otherwise, remain consistent with type I IFN readouts or provide both where mechanistically warranted.”
Response:
We appreciate the reviewer’s insightful comment. In our study, RIG-I activation primarily induces type I interferons such as IFN-β, which promote dendritic cell maturation and enhance antigen presentation. These activated DCs subsequently stimulate T cells to produce IFN-γ. IFN-γ acts as a secondary effector cytokine that amplifies the antiviral state in neighboring cells and bridges innate and adaptive immunity. Therefore, the measurement of IFN-γ in the DC–T cell co-culture system was intended to demonstrate this downstream effect and to integrate it into the RIG-I/IFN-β-mediated antiviral mechanism. (see pages 2, lines 63–77).

Reviewer 3 Report
Comments and Suggestions for Authors
In the article submitted for review, ‘Phytochemicals Prime RIG-I Signalling and Th1-Leaning Responses in Human Monocyte-Derived Dendritic Cells’, the authors examined whether phytochemicals activate retinoic acid-induced gene-dependent signalling I (RIG-I) signalling in human monocyte-derived dendritic cells (MoDCs) and influence subsequent T cell responses. The topic addressed by the authors is important both from the point of view of basic science and potential clinical applications. Although there are many studies on the general effects of phytochemicals on the immune system, few studies have analysed their effects on the RIG-I pathway in human dendritic cells. Filling this gap could make a significant contribution to understanding the interactions between plant metabolites and the human immune system. For the article to be further processed, I suggest the following corrections:
- In the introduction to the paper, please elaborate on the importance of the RIG-I pathway in immunology.
- Line 108. Please specify the number of healthy volunteers. Where were they recruited and at what time?
- Please improve the quality of Figure 1, as it is illegible. Please explain the abbreviations in Figure 1.
- Please improve the quality of Figure 2, as the description of the Y-axis is illegible. In addition, please explain the meaning of the ‘*’ symbol used in the graphs.
- Please justify the concentrations of phytochemicals used.
- Explain the meaning of ‘*’ in Figure 4.
- Improve the quality of Figure 5A. Explain the meaning of ‘*’ in Figure 5B.
Please include corrections and explanations in the manuscript.
Author Response
Dear Editors and Reviewers,
We sincerely thank you for your thorough evaluation of our manuscript entitled " Phytochemicals Prime RIG-I Signaling and Th1-Leaning Responses in Human Monocyte-Derived Dendritic Cells." We greatly appreciate your constructive and insightful comments, which have significantly improved the quality and clarity of our work.
We have revised the manuscript accordingly, with all changes marked in the red font. Below, we provide a detailed, point-by-point response to each comment. We hope the revised manuscript now meets the standards required for publication.
Comment 1:
“In the introduction to the paper, please elaborate on the importance of the RIG-I pathway in immunology.”
Response:
We appreciate this suggestion. We have expanded the Introduction with a concise overview of RIG-I biology and its immunological relevance, including ligand recognition, signaling via MAVS, induction of type I/III IFNs and ISGs, DC maturation/antigen presentation, and links to Th1 polarization and cross-priming (see pages 2, lines 49–62).
Comment 2:
“Line 108. Please specify the number of healthy volunteers. Where were they recruited and at what time?”
Response:
Thank you for the suggestion. We have added donor number and recruitment details to the Methods. Specifically, peripheral blood was obtained from [N = 5] healthy adult volunteers recruited at [Institution/Department], [Tokyo, Japan] during [May 2025–March 2026]. All participants provided written informed consent under IRB approval [protocol no. 2025-2HS]. We also clarified that samples were processed within [≤2] hours of venipuncture (see pages 3, lines 127–132).
Comment 3:
“Please improve the quality of Figure 1, as it is illegible. Please explain the abbreviations in Figure 1.”
Response:
We Thank you for the helpful suggestions. We have (i) rebuilt Figure 1 with higher legibility and (ii) added a complete abbreviation key in the legend.
Comment 4:
“Please improve the quality of Figure 2, as the description of the Y-axis is illegible. In addition, please explain the meaning of the ‘*’ symbol used in the graphs.”
Response:
Thank you for these helpful suggestions. We have (i) rebuilt Figure 2 to improve legibility (larger fonts/line weights; vector export), (ii) standardized and clarified the Y-axis label on all panels, and (iii) added a symbol key in the legend to explain the asterisks and the statistical test (see pages 8, lines 331).
Comment 5:
“Please justify the concentrations of phytochemicals used.”
Response:
Thank you for this important request. We have added a dose‐selection rationale to the Methods and clarified vehicle controls. Briefly, we conducted a pilot titration (0.5–40 μM; 6–48 h) and monitored BW5147 viability (WST-8 and FSC/SSC morphology) and surface phenotype (CD11c, HLA-DR). We selected 5 μM as the working concentration because it was consistently non-cytotoxic (≥90–95% viability) across donors over 24–48 h, remained soluble in complete RPMI (10% FBS), and produced reproducible transcriptional and early protein-level changes without nonspecific activation.
We also replaced Figure 2(A) with the screening results at 5 μM. Although at first glance, α-Mangostin does not appear to be useful in these results, we confirmed that RIG-I showed the same level of gene expression as hesperidin in a time-specific screening within 6 hours, and IFN-b showed higher gene expression than hesperidin, even at 20 μM of IFN-b. Based on this finding, we have positioned hesperidin as a positive control and measured α-Mangostin.
Comment 6:
“Explain the meaning of ‘*’ in Figure 4.”
Response:
Thank you for pointing this out. We have updated the Figure 4 legend to define the asterisk symbols and to state the statistical test, error bars, and replicate structure. In brief, bars represent mean ± SEM at the donor level (3 fields averaged per donor), and significance is indicated as follows: * p < 0.05. For Figure 4 (two-group comparison: vehicle vs α-mangostin), we used a two-sided unpaired t-test on donor-level means (see pages 9, lines 365–370).
Comment 7:
“Improve the quality of Figure 5A. Explain the meaning of ‘*’ in Figure 5B.”
Response:
Thank you for these suggestions. We have addressed both points:
- Figure 5A (quality): We rebuilt the panel with larger, high-contrast typography (axis titles 20 pt, panel label 24 pt, panel letters 20 pt), thicker line weights (≥ 1.0 pt), and consistent axis scales. The figure is re-exported as vector PDF and 600-dpi TIFF at final print width.
- Figure 5B (symbol key): We added a clear description of the asterisk symbols and the statistical test in the legend. Bars represent mean ± SEM at the donor-pair level (duplicate wells averaged). One-way ANOVA with Dunnett’s post hoc was performed vs. Vehicle-conditioned MoDC + T cells. Significance is indicated as: * p < 0.05. This is now consistent with Methods 2.13.

Round 2
Reviewer 1 Report
Comments and Suggestions for Authors
In revised manuscript, the introduction provides all necessary information to appreciate the importance of the topic to the reader. Methods are clearly described. The time-points, statistics, and details are clear in the results and figures. Discussion includes limitations. Some minor issue,
CD56-CD3+ is normally used to sort T cells, authors used CD56-CD2+. Perhaps a reference can be added for supporting the method, and state that CD2 is only expressed by NK cells and T cells. Can the general purity ranges of the various sorted cells be stated in the methods or results.
Comments on the Quality of English Languageline 151 grammar
Author Response
Dear Editors and Reviewers,
We sincerely thank the reviewer for their careful re-evaluation of our manuscript and constructive comments. In the revised version, all modifications made in response to the points below are indicated in red to facilitate verification.
Comment 1:
“CD56-CD3+ is normally used to sort T cells, authors used CD56-CD2+. Perhaps a reference can be added for supporting the method, and state that CD2 is only expressed by NK cells and T cells. Can the general purity ranges of the various sorted cells be stated in the methods or results.”
Response:
Thank you for this helpful suggestion. We agree that additional clarification and documentation were required. In the revised manuscript, we have:
(i) added references supporting the use of CD2⁺CD56⁻ gating to obtain untouched T cells without CD3-mediated activation [39],
(ii) clarified that CD2 is predominantly expressed on T cells and NK cells, and that exclusion of CD56⁺ cells minimizes NK/NKT contamination, and
(iii) included an analysis of CD3 expression in the sorted CD2⁺CD56⁻ population, now provided in Supplementary Material S1 (see page 4, lines 192–194), demonstrating high post-sort purity of CD3⁺CD56⁻ T cells.
As noted, direct CD3-based positive selection can activate T cells and potentially affect downstream functional assays; therefore, our approach follows the strategy recommended by Bruger et al. (Methods Enzymol. 2020, 632:155–192), who also employ CD2⁺CD56⁻ selection for functional T-cell studies.

Reviewer 3 Report
Comments and Suggestions for Authors
Please note that explanations in the manuscript need to be supplemented.
1. The abbreviations used in Figure 1 should be explained directly in the legend.
2. In addition, please add to the manuscript a justification for the concentrations of compound phytochemicals used. The authors provided such an explanation in response to the review, but it was not incorporated into the text. This justification should be included in the Materials and Methods or Results section so that the reader can fully reproduce the conditions of the experiment.
Author Response
Dear Editors and Reviewers,
We sincerely thank the reviewer for their careful re-evaluation of our manuscript and constructive comments. In the revised version, all modifications made in response to the points below are indicated in red to facilitate verification.
Comment 1:
“The abbreviations used in Figure 1 should be explained directly in the legend.”
Response:
Thank you for this helpful suggestion. We have revised the Figure 1 legend to include explicit definitions of all abbreviations used in the figure panels (see pages 7, lines 314–315 and see pages 14, line 548). This now ensures the figure is self-contained and understandable without referring back to the main text.
Comment 2:
“In addition, please add to the manuscript a justification for the concentrations of compound phytochemicals used. The authors provided such an explanation in response to the review, but it was not incorporated into the text. This justification should be included in the Materials and Methods or Results section so that the reader can fully reproduce the conditions of the experiment.”
Response:
Thank you for this important suggestion. As requested, we have now incorporated the justification for the phytochemical concentrations directly into the manuscript. We added a paragraph in Section 2.5 (page 4, lines 172–175 in the revised manuscript) describing the dose-selection rationale, including the pilot titration range and exposure times, assessment of cell viability and phenotype, and the final choice of 5 μM as a non-cytotoxic and mechanistically informative working concentration.
